

# Deep-Time Marine Sedimentary Element Database

**Jiankang Lai[1], Haijun Song[1*], Daoliang Chu[1], Jacopo Dal Corso[1], Erik A. Sperling[2], Yuyang Wu[1*], Xiaokang Liu[1], Lai Wei[3], Mingtao Li[4], Hanchen Song[1], Yong Du[1], Enhao Jia[1], Yan Feng[1], Huyue Song[1], Wenchao Yu[1], Qingzhong Liang[5], Xinchuan Li[5], Hong Yao[5]**

[1]State Key Laboratory of Biogeology and Environmental Geology, School of Earth Sciences, China University of Geosciences, Wuhan 430074, China

[2]Department of Earth and Planetary Sciences, Stanford University, Stanford, CA 94305, USA.

[3]School of Future Technology, China University of Geosciences, Wuhan 430074, China

[4]School of Resources and Environment, Linyi University, Linyi 276000, China

[5]School of Computer Science, China University of Geosciences, Wuhan 430074, China

Corresponding authors: Haijun Song (haijunsong@cug.edu.cn), Yuyang Wu (wuyuyang@cug.edu.cn)

**Abstract.** Geochemical data from ancient marine sediments are crucial for studying palaeoenvironments, palaeoclimates, and elements' cycles. With increased accessibility to geochemical data, many databases have emerged. However, there remains a need for a more comprehensive database that focuses on deep-time marine sediment records. Here, we introduce the "Deep-Time Marine Sedimentary Element Database" (DM-SED). The DM-SED has been built upon the "Sedimentary Geochemistry and Paleoenvironments Project" (SGP) database with the new compilation of 34,938 data entries from 433 studies, totalling 63,691 entries. The DM-SED contains 2,412,085 discrete marine sedimentary data points, including major and trace elements and some isotopes. It includes 9,271 entries from the Precambrian and 54,420 entries from the Phanerozoic, thus providing significant references for reconstructing deep-time Earth system evolution. The data files described in this paper are available at https://doi.org/10.5281/zenodo.13898366 (Lai et al., 2024).





## 1 Introduction

Geochemical data from deep-time marine sediments are fundamental for reconstructing the evolution of the Earth system. By analysing the concentrations of chemical elements in sediments and their isotopic compositions, we can reconstruct the past cycling of elements in the Earth's surface systems and reveal its evolution through time (Large et al., 2015; Reinhard et al., 2017; Farrell et al., 2021; Planavsky et al., 2023). For instance, total organic carbon (TOC), phosphorus (P), biogenic barium ($Ba_{bio}$), copper (Cu), zinc (Zn), nickel (Ni), etc., enable reconstruct marine primary productivity and carbon cycle changes, thereby revealing past climate change mechanisms (Scott et al., 2013; Schoepfer et al., 2015; Shen et al., 2015; Schoepfer et al., 2016; Xiang et al., 2018; Jin et al., 2020; Tribovillard, 2021; Wang et al., 2022; Zhang et al., 2022; Li et al., 2023; Sweere et al., 2023; Zhao et al., 2023). Elements such as uranium (U), vanadium (V), and molybdenum (Mo) can reveal how marine redox conditions changed during critical periods in animal evolution, including mass extinctions and evolutionary radiations (Algeo and Liu, 2020; Schobben et al., 2020; Stockey et al., 2024). Oxygen isotopes ($\delta^{18}O$) in the remains of marine fossil animals can reveal oceanic palaeo-temperature changes (Veizer and Prokoph, 2015; Song et al., 2019; Grossman and Joachimski, 2020; Scotese et al., 2021; Judd et al., 2022). However, many geochemical studies focused on high-resolution research of limited time intervals and/or regions, and there is little comprehensive exploration across large-scale geological time and globally.

Fortunately, with more journals and institutions adopting strict data archiving rules and promoting adherence to FAIR (Findability, Accessibility, Interoperability, and Reusability) principles (Wilkinson et al., 2016; "FAIR Play in Geoscience Data," 2019), a large amount of geochemical data has become accessible, and sample meta-data records are more detailed. Several geochemical databases of varying scales and foci have emerged, such as the following:

- EarthChem, which covers igneous, sedimentary, and metamorphic rocks and comprises numerous joint databases (https://www.earthchem.org/, last accessed: 16 July 2024).

- Petrological Database of the Ocean Floor (PetDB), which includes elemental chemical, isotopic, and mineralogical data of global ocean floor igneous rocks, metamorphic rocks, minerals, and inclusions (https://www.earthchem.org/petdb, last accessed: 16 July 2024).

- Geochemistry of Rocks of the Oceans and Continents (GEOROC), a comprehensive compilation



of chemical, isotopic, and other data on igneous rock samples, including whole rock, glass, mineral,
and inclusion analyses and metadata (http://georoc.mpch-mainz.gwdg.de, last access: 16 July

61      2024).

●   Data Publisher for Earth & Environmental Science (PANGAEA), which is used for archiving,
publishing, and disseminating georeferenced data from earth, environmental, and biodiversity
sciences and includes a large number of sediment core data (https://www.pangaea.de, last accessed:
16 July 2024).
●   Stable Isotope Database for Earth System Research (StabisoDB) containing $\delta^{18}O$ and $\delta^{13}C$ data for
more than 67,000 macrofossil and microfossil samples, including benthic and planktonic
foraminifera, benthic and nektonic mollusks, brachiopods, fish teeth, and conodonts
(https://cnidaria.nat.uni-erlangen.de/stabisodb/, last accessed: 16 July 2024).
●   Sedimentary Geochemistry and Paleoenvironments Project (SGP), which collects multi-proxy
sedimentary geochemical data with an emphasis on Neoproterozoic-Palaeozoic shale data in its
first data release (https://sgp-search.io/, last accessed: 12 June 2024).
Many other government initiatives also host databases:
●   The United States Geological Survey (USGS) National Geochemical Database, an archive of
geochemical    information    and    related    metadata    from    USGS    research
(https://www.usgs.gov/energy-and-minerals/mineral-resources-program/science/national-geochemi
cal-database, last accessed: 16 July 2024).
●   The British Geological Survey (BGS), which provides data and information on UK geology,
boreholes, geomagnetism, groundwater, rocks, etc. (http://www.bgs.ac.uk/, last accessed: 16 July

80      2024).

●   The Australian National Whole Rock Geochemistry Database (OZCHEM), including chemical
compositions of rock, soil, and sediment samples (https://ecat.ga.gov.au/geonetwork/srv/, last
accessed: 16 July 2024).
Although some of these databases (Table 1) include data on ancient marine sediments, they are
often limited to specific countries or regions and have certain shortcomings, such as the lack of age
data, the absence of many recent publications, missing information from original individual
publications, and relatively coarse age resolutions. Thus, we have established the Deep-Time Marine
Sedimentary Element Database (DM-SED), which focuses on the elemental content changes in marine





sediments across geological history. The current version of the DM-SED database contains 63,691
entries, enabling research on a series of scientific issues related to palaeoenvironmental, palaeoclimatic,
and elemental cycles in deep-time Earth history.
**Table 1. Overview of different databases (Note: not all databases have a clear number of records).**

| Database name | Content | Website information | Number of records | Data regions |
|---|---|---|---|---|
| EarthChem | Igneous, sedimentary, and metamorphic rocks; various joint databases | https://www.earthchem.org/, last accessed: 16 July 2024 | Over 2,596 digital content files in EarthChem Library | Global |
| PetDB | Elemental chemical, isotopic, and mineralogical data of global ocean floor rocks | https://www.earthchem.org/petdb, last accessed: 16 July 2024 | over 6,000,000 samples | Global |
| GEOROC | Chemical, isotopic, and other data on igneous rock samples | http://georoc.mpch-mainz.gwdg.de, last access: 16 July 2024 | 672,990 samples | Global |
| PANGAEA | Georeferenced data from earth, environmental, and biodiversity sciences | https://www.pangaea.de, last accessed: 16 July 2024 | Extensive dataset | Global |
| StabisoDB | $\delta^{18}O$ and $\delta^{13}C$ data for macrofossil and microfossil samples | https://cnidaria.nat.uni-erlangen.de/stabisodb/, last accessed: 16 July 2024 | Over 67,000 samples | Global |
| SGP | Multi-proxy sedimentary geochemical data from the Palaeozoic and Neoproterozoic | https://sgp-search.io/, last accessed: 12 June 2024 | 82,578 samples | Global |
| USGS | Geochemical information and related metadata from USGS research | https://www.usgs.gov/energy-and-minerals/mineral-resources-program/science/national-geochemical-database, last accessed: 16 July 2024 | Extensive dataset | United States |
| BGS | Data on UK geology, boreholes, geomagnetism, groundwater, rocks, etc. | http://www.bgs.ac.uk/, last accessed: 16 July 2024 | Extensive dataset | United Kingdom |
| OZCHEM | Chemical compositions of rock, soil, and sediment samples | https://ecat.ga.gov.au/geonetwork/srv/, last accessed: 16 July 2024 | Extensive dataset | Australia |




DM-SED version 0.0.1 is presented in table (.csv) format. Dynamic versions of the most recent
release can be found on Zenodo (https://doi.org/10.5281/zenodo.13898366, last accessed: 7 October
2024) (Lai et al., 2024), and a static copy of Version 0.0.1 is archived in the Geobiology data
(http://202.114.198.132/dmgeo-geobiology-portal/, last accessed: 25 September 2024). In the following
sections, we provide a brief overview of the database, information on the data sources and selection
criteria, and a review of the definitions and decisions behind the metadata fields associated with each
proxy measurement. We explore the spatial and temporal distribution trends of the compiled data and
discuss future uses and limitations of the database.

**2 Dataset overview**
The DM-SED aims at collecting geochemical data from deep-time marine sediments. It is
primarily based on the SGP database, but with 34,938 entries of new compiled data. The SGP has a
total of 82,578 entries, we selected only 28,753 entries on marine sedimentary geochemical data, and is
comprised of three parts: two parts from the U.S. Geological Survey (USGS), i.e. the National
Geochemical Database (USGS NGDB, https://mrdata.usgs.gov/ngdb/rock, last accessed: 9 September
2024) and the Global Geochemical Database for Critical Metals in Black Shales project (USGS
CMIBS, Granitto et al., 2017), with samples mainly from North America and Phanerozoic shales from
various continents, respectively (Farrell et al., 2021). The third part comprises direct inputs by SGP
members. The direct inputs in the Phase 1 SGP data release primarily focused on
Neoproterozoic–Palaeozoic shales, although there are other lithologies and other time periods
represented (Farrell et al., 2021). Our DM-SED database, built upon the SGP, includes a new
compilation of 34,938 entries from 433 literatures, covering a time range from approximately 3800 Ma
to the present, and including entries from North America, Europe, Asia, Africa, South America,
Oceania, Pacific and Atlantic, thus supplementing the temporal and spatial distribution gaps in the SGP
database and thereby creating a more comprehensive sedimentary marine geochemical database. The
new compiled literatures span the time range from 1965 to 2023, with the number of papers per decade
gradually increasing (Fig. 1). It should be noted that the top of the DM-SED version 0.0.1 data is the
new compilation, and the bottom contains data imported from SGP.



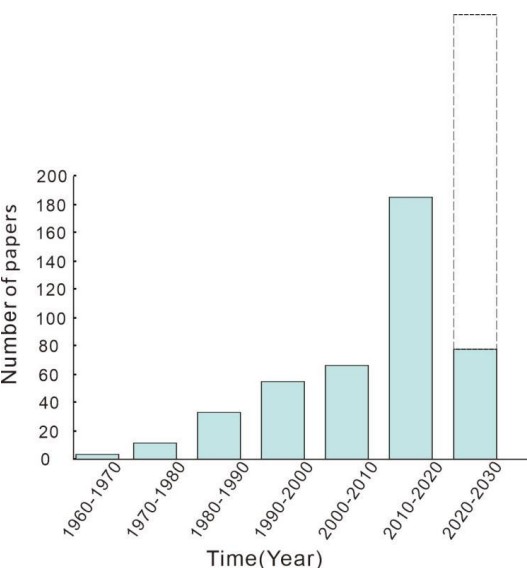


**Figure 1. The distribution of publication years for newly compiled literature (the dashed line denotes the**

**predicted literature from 2023 to 2030).**


**Table 2. Summary of data entries and points in the DM-SED.**

|  | Entries | Data points |
|---|---|---|
| **New compilation** | 34,938 | 1,345,589 |
| **SGP** | 28,753 | 1,066,496 |
| **DM-SED** | 63,691 | 2,412,085 |

The DM-SED database comprises 63,691 entries with 2,412,085 discrete data points (Table 2),
each including location (SampleID, SampleName, SiteName, Region, Elevation, SampleDepth,
ModLat, ModLon, PalaeoLat, PalaeoLon), age (Age, Period, Stage, Biozone), stratigraphic information
(LithName, LithType, Formation, Facies), carbon element (total carbon (Total C), inorganic carbon
($C_{inorg}$), TOC, in wt%, isotopic values ($\delta^{18}O_{carb}$, $\delta^{13}C_{Ker}$, $\delta^{13}C_{TOC}$, $\delta^{13}C_{carb}$, $\delta^{34}S_{CAS}$, $\delta^{34}S_{pyr}$, $\delta^{15}N_{total}$,
$\delta^{15}N_{org}$, in ‰), major element (P, Al, Si, Ti, Fe, Ca, Mg, Na, K, S, N, in wt%), trace element (Ag, Ar,
As, B, Ba, Be, Bi, Br, Cd, Ce, Co, Cr, Cs, Cu, Dy, Er, Eu, Ga, Gd, Ge, Hf, Hg, Ho, In, La, Li, Lu, Mn,
Mo, Nb, Nd, Ni, Pb, Pr, Rb, Re, Sb, Sc, Se, Sm, Sn, Sr, Ta, Tb, Te, Th, Tl, Tm, U, V, W, Y, Yb, Zn, Zr,
in ppm), and data sources (Reference, Project). The specific names and descriptions of each field in the
database are shown in Table 3. The standards and descriptions of isotope ratios in the database are





shown in Table 4.
**Table 3. Field names and descriptions.**

| Field name | Description of field (units) |
|---|---|
| ***Location fields*** | |
| SampleID | Unique sample identification code |
| SampleName | Author denoted title for the sample (often non-unique) |
| SiteName | Name of the drill core site or section |
| Region | Country or ocean of the data collection site |
| Elevation | Distance between sampling location and sea level (m) |
| SampleDepth | Stratigraphic height or depth (m) |
| ModLat | Modern latitude of collection site rounded to two decimals; negative values indicate the Southern Hemisphere (decimal degrees) |
| ModLon | Modern longitude of the collection site rounded to two decimals; negative values indicate the Western Hemisphere (decimal degrees) |
| PalaeoLat | Palaeolatitude of collection site rounded to two decimals; negative values indicate the Southern Hemisphere (decimal degrees) |
| PalaeoLon | Palaeolongitude of the collection site rounded to two decimals; negative values indicate the Western Hemisphere (decimal degrees) |
| ***Age fields*** | |
| Age | Absolute Age, in reference to GTS2020 (Ma) |
| Period | The geologic period |
| Stage | The geologic stage (i.e. geochronologic age) |
| Biozone | Conodont, graptolite, ammonite biozone, etc |
| ***Stratigraphy*** | |
| LithName | Lithological name of the sample, as originally published |
| LithType | Lithology type of sample (e.g. carbonate, siliciclastic) |
| Formation | Geologic formation name |
| Facies | Depositional environment (e.g. mid-shelf, ramp) |
| ***Proxy fields*** | |
| Carbon | The content of carbon, including Total C, $C_{inorg}$, TOC, rounded to two decimals (wt%) |
| Isotopes | The isotope value, rounded to two decimals (‰) |
| Major elements | The content of major elements such as P, Al, and Si, rounded to two decimals (wt%) |
| Trace elements | The content of trace elements such as Ag, Ar, As, B, and Ba, rounded to two decimals (ppm) |
| ***Data sources*** | |
| Reference | Data sources, including published literature or other databases |
| Project | Two parts: new compilation and SGP |






**Table 4. Standards and descriptions of isotope ratios in the DM-SED.**

| Symbol | Standard | Description |
|---|---|---|
| $\delta^{18}O_{carb}$ | Vienna Pee Dee Belemnite (VPDB) | Oxygen isotope ratio of carbonate minerals, used in palaeoclimate studies. |
| $\delta^{13}C_{Ker}$ | VPDB | Carbon isotope ratio of kerogen, used to study the source and depositional environment of organic matter. |
| $\delta^{13}C_{TOC}$ | VPDB | Carbon isotope ratio of total organic carbon, used to analyse the source of organic matter and biogeochemical cycles in sediments. |
| $\delta^{13}C_{carb}$ | VPDB | Carbon isotope ratio of carbonate minerals, used in palaeoclimate and carbon cycle research. |
| $\delta^{34}S_{CAS}$ | Vienna Canyon Diablo Troilite (VCDT) | Sulfur isotope ratio of carbonate-associated sulfate, used to study the sulfur cycle and redox conditions. |
| $\delta^{34}S_{pyr}$ | VCDT | Sulfur isotope ratio of pyrite, typically used to investigate the sulfur cycle and redox conditions in ancient oceans. |
| $\delta^{15}N_{total}$ | Atmospheric Nitrogen (air $N_2$) | Nitrogen isotope ratio of total nitrogen, used to study the nitrogen cycle and nutrient sources. |
| $\delta^{15}N_{org}$ | air $N_2$ | Nitrogen isotope ratio of organic nitrogen, often used to analyse the source of organic matter and the nitrogen cycle. |


## 3 Dataset screening and processing


This section details the screening and processing criteria for sample location, age, lithology and facies,
specific geochemical values, and data source information (Fig. 2).



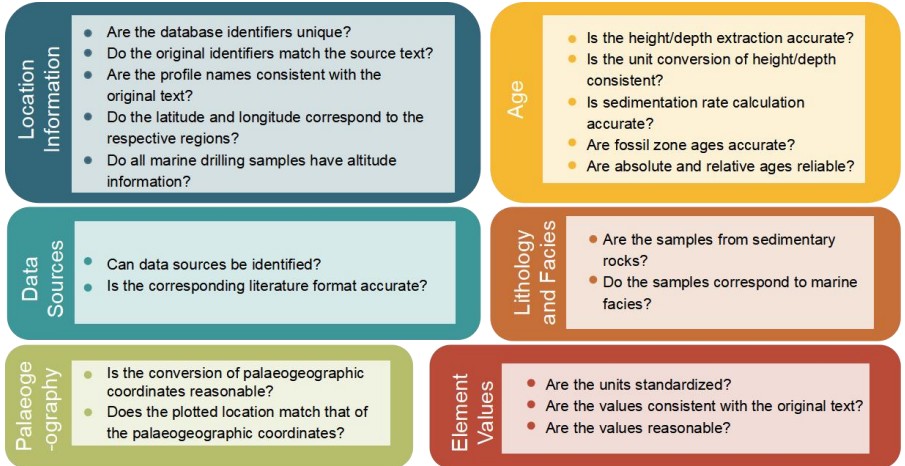

**Figure 2. The data filtering and processing criteria for DM-SED.**

For sample location, the dataset includes SampleID, SampleName, SiteName, Region, Elevation, SampleDepth, ModLat, ModLon, PalaeoLat, and PalaeoLon. A unique SampleID is assigned to each sample in the DM-SED. The SampleName corresponds to the identifier given in each referenced publication, facilitating cross-referencing with the original data. The SiteName includes well name or outcrop information, representing the smallest unit of location information. The Region indicates the country or ocean area where the sample has been collected and represents a broader geographical range. The Elevation data are mainly related to samples from the Deep Sea Drilling Project (DSDP) and the Ocean Drilling Program (ODP) collected from post-Cretaceous sediments and indicate whether the samples originate from deep or shallow marine environments. SampleDepth refers to the relative position (in metres) of the sample within the well or outcrop, which is crucial for calculating sample age. In some publications, specific heights are not provided directly but are given as relative heights through figures. We manually extracted these heights using WebPlotDigitizer, rounding to two decimal places (Drevon et al., 2017). For publications in which heights are expressed in feet or centimetres, we converted the units to metres. Modern latitude and longitude (ModLat and ModLon) information are the most precise location data. Although some publications provide exact coordinates, many offer only section names (i.e. SiteName) and regions or merely a map marking the location of the section. For publications providing section names, we determined accurate coordinates by consulting other studies carried out in the same section. For those providing only a map marking the location of the section, we used Google Maps to estimate relative coordinates. To ensure consistency, we recorded sample





coordinates in decimal degrees, rounded to two decimal places, with positive values indicating north
latitude and east longitude and negative values indicating south latitude and west longitude. For
palaeo-coordinates, we reconstructed palaeo-latitude and palaeo-longitude (PalaeoLat, and PalaeoLon)
using the sample age and modern coordinates, employing the PointTracker v7 rotation files from the
PALEOMAP project, which are based on current geographic reference data and global tectonic history
models (Scotese, 2008). It is important to note that we only generated palaeogeographic locations for
samples from the Phanerozoic, as the geological records from this time are more complete and
abundant compared to those from the Precambrian, making the reconstruction of geographic features
(such as ancient oceans, mountains, plains, etc.) relatively more reliable and accurate (Scotese and
Wright 2018). We plotted the sample points on palaeogeographic maps based on Scotese's data using
QGIS 3.16 (Scotese and Wright 2018).

To assign specific ages to each sample in the database, we assumed a constant sedimentation rate

within the same formation or group of section. If the original studies provided numerical ages for two
or more samples, we calculated the precise age for each sample based on the sedimentation rate and
assigned it accordingly. If absolute ages were not provided in the original literature, we assigned
approximate ages based on corresponding fossil zones or the general age of the same lithostratigraphic
unit in the same region (Farrell et al., 2021; Judd et al., 2022). For samples with completely missing
height information in the original text, we assigned the same age to all samples within the section based
on lithostratigraphic information (mainly samples from USGS NGDB and USGS CMIBS). Once each
sample had a specific age, we assigned it to a specific Period and Stage according to its age. We
attempted to incorporate the most recent age models; however, due to the extensive size of the data
compilation, it was not feasible to update all of them. All ages were based on the timescale provided by
the Geologic Time Scale 2020 (Gradstein et al., 2020).

For lithology and facies, the lithologies include shale, mudstone, sandstone, limestone, dolomite,

and others. We classified these into two major types of rock: siliciclastic sedimentary rocks (88.7%)
and carbonate rocks (11.3%). For outcrop sections, lithostratigraphic unit was generally available;
however, for marine drilling data, there were no corresponding group names. Regarding facies
classification, before the Cretaceous, the primary depositional environment was marine settings on
continental crust, including specific facies such as tidal flats, inner shelves, outer shelves, and basinal.
However, after the Cretaceous, with most samples coming from DSDP and ODP, deep ocean



depositional environments emerged.
For specific geochemical values in the DM-SED database, we standardized the units, converting
oxides to elements (e.g. P (ppm) to P (wt%), $P_2O_5$ (wt%) to P (wt%)). If a sample was analysed
multiple times, we averaged the value. For literature before 2000, some data were preserved as images,
requiring manual extraction of values, and some images were slightly blurry, potentially leading to
minor human error. We excluded data that were beyond detection limits (e.g. the trace element content
is too low and the value provided in the text represents the minimum detection limit) or unreasonable
(e.g. negative values for major and trace elements).
Regarding data sources, we ensured that each corresponding reference was collected and listed in
full citation format, including authors, title, publication date, journal, page numbers, and DOI. Most
data in the SGP database came directly from USGS NGDB and USGS CMIBS, without corresponding
literature sources, so we marked them individually. And the project includes two parts: new
compilation and SGP. We used keyword searches in Google Scholar to identify missing references and
made efforts to target literature for data-scarce regions (e.g. South America) and time intervals (e.g.
Silurian, Jurassic).
**4 Data distribution**
The elemental data content distribution for the entire database is shown in Fig. 3. Overall, major
elements have the highest data quantity, followed by trace elements and carbon elements, with isotope
data having the lowest quantity. Among the major elements, N has the fewest entries, with 3,164
records, whereas the other major elements all have more than 10,000 entries. Al has the highest
quantity, with 50,568 records. Among trace elements, Mn has the largest record (41,058 records),
followed by Ba (40,163 records). Ar and Br have the fewest records, with 9 and 162 records,
respectively. Other elements such as Ag, B, Bi, Ge, Hg, Ho, In, Pr, Re, Sb, Se, Sn, Ta, Te, Tl, Tm, and
W have data quantities ranging from 1,000 to 10,000. Elements such as As, Be, Cd, Ce, Co, Cr, Cs, Cu,
Dy, Er, Eu, Ga, Gd, Hf, La, Li, Lu, Mo, Nb, Nd, Ni, Pb, Rb, Sc, Sm, Sr, Tb, Th, U, V, Y, Yb, Zn, and
Zr all have more than 10,000 records each. For carbon elements, TOC has the most records, with
33,216 entries, followed by Total C with 9,201 entries. $C_{inorg}$ has the fewest records, with 7,194 entries.
Isotope data are overall less abundant, with none exceeding 10,000 entries; the most abundant is

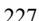

$\delta^{13}C_{TOC}$, with 8,166 records, and the least abundant is $\delta^{13}C_{Ker}$, with only 29 records.

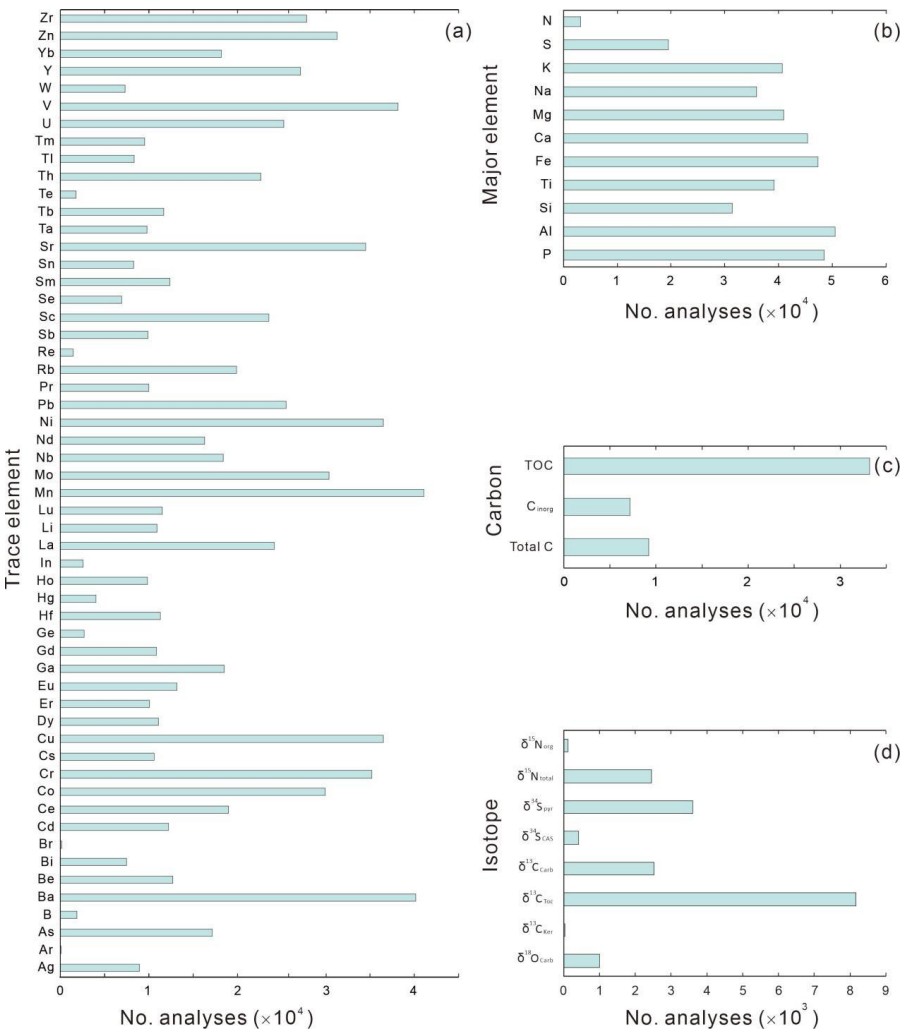


**Figure 3. Histogram distribution of different subsets. (a) Trace elements. (b) Major elements. (c) Carbon**
**elements. (d) Isotopes.**
The temporal trend of data density in the entire database, shown in Fig. 4a, indicates that the data
are primarily distributed in the Phanerozoic Eon, which accounts for 85% of the entire database. From
this, the Cenozoic Era accounts for 19% of the database, the Mesozoic Era accounts for 21%, and the
Palaeozoic Era accounts for 45%. Precambrian data account for only 15% of the entire database. The
SGP data are most concentrated in the Palaeozoic Era, in which they make up 27% of the total database,





with the new compiled data contributing only 18%. In other eras, the new compiled data outnumber the
SGP data: 4% versus 15% in the Cenozoic, 7% versus 14% in the Mesozoic, and 7% versus 8% in the
Precambrian. This is mainly the case because the SGP data in the first phase were primarily from the
Neoproterozoic and Palaeozoic eras (Farrell et al., 2021).

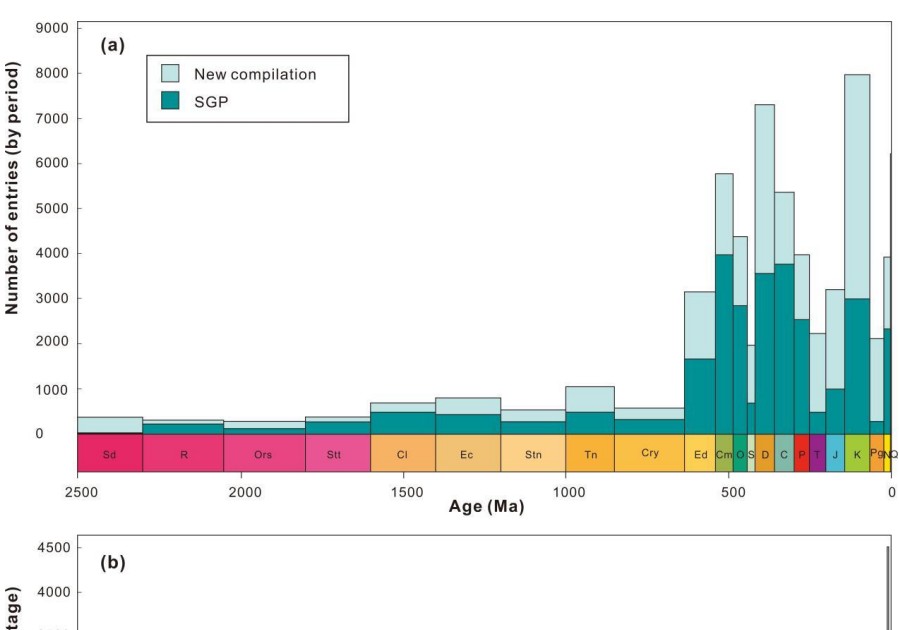

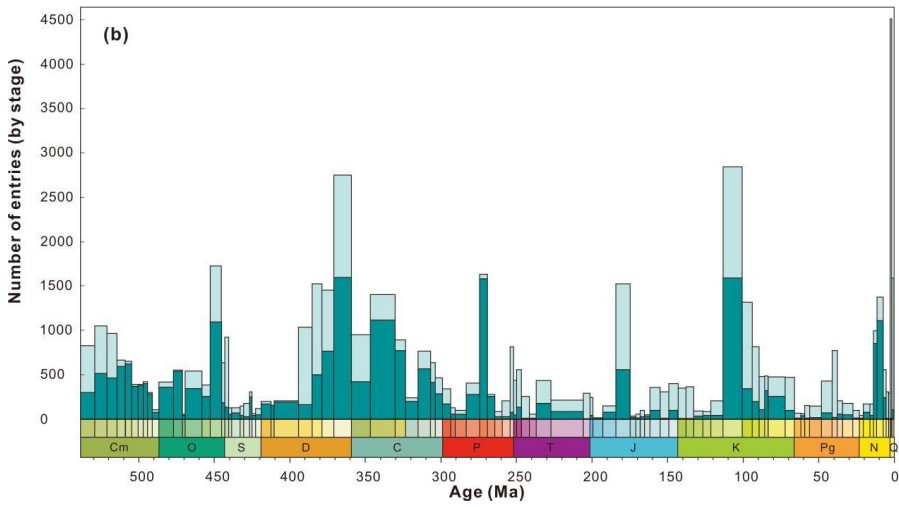


**Figure 4. The age distribution of samples in the database. (a) Age distribution of samples (excluding a small**
**number of samples with ages >2500 Ma from the figure, a total of 1298 samples). (b) Age distribution of**
**Phanerozoic samples at the stage level.**


For the distribution of sample ages within the Phanerozoic, we divided the samples by stage, as
shown in Fig. 4b. For the Quaternary Period, due to its short duration, data were not subdivided by
Stage, but only into Holocene and Pleistocene. Data distribution is not uniform, with the highest
concentration in the Quaternary Period. These data mainly come from DSDP and ODP, which are
characterised by a high number of core samples and high resolution. There are fewer data for the Upper
Permian, Lower Triassic, and Lower to Middle Jurassic, possibly because of the existence of Pangaea
at that time, which reduced the area of continental margins and inhibited marine transgressions,
resulting in fewer preserved marine environments in comparison to those of other geological periods
(Mackenzie and Pigott, 1981; Walker et al., 2002). The distribution of sample quantities in other
periods fluctuates, often corresponding to periods of significant research interest, such as the
end-Ordovician, end-Devonian, end-Permian, Early Jurassic Toarcian and Early Cretaceous Albian,
which had peaks in sample numbers due to their association with major mass extinction events and
oceanic anoxic events (Fan et al., 2020).

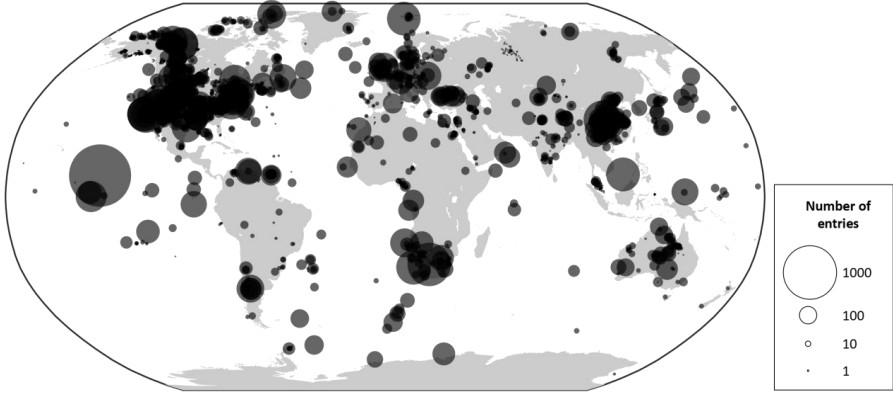


**Figure 5. Bubble chart of modern geographical distribution and sample quantities in the database.**
In terms of spatial trends, the spatial distribution of sampling points in the DM-SED database is
inherently uneven, both in modern and palaeogeographic locations. Modern locations are primarily
concentrated in North America, Europe, South Africa, and China (Fig. 5). When modern coordinates
are converted to palaeogeographic coordinates and projected onto palaeogeographic maps, Cambrian to
Jurassic data come predominantly from continental margin environments, as oceanic crust plates
subducting before the Cretaceous led to preservation of very few deep-sea environments (Fig. 6).
Cambrian and Ordovician data are distributed mainly on the Laurentia, Baltica, and South China plates,



with a few along the Gondwana margin. Silurian data occur mainly on Laurentia, South China, and
right side of Gondwana. Devonian and Carboniferous data are primarily on the Laurussia plate, with
sparse distribution in South China and Gondwana. Permian and Triassic data are mainly on the
Laurussia and South China plates, with sparse distribution in Gondwana. Jurassic data are primarily on
the North American, European shelf, with sparse distribution on other plates. From the Cretaceous to
the Quaternary, sample locations, dominated by data from the DSDP, ODP, and USGS NGDB projects,
are mainly in the deep oceans and North America.

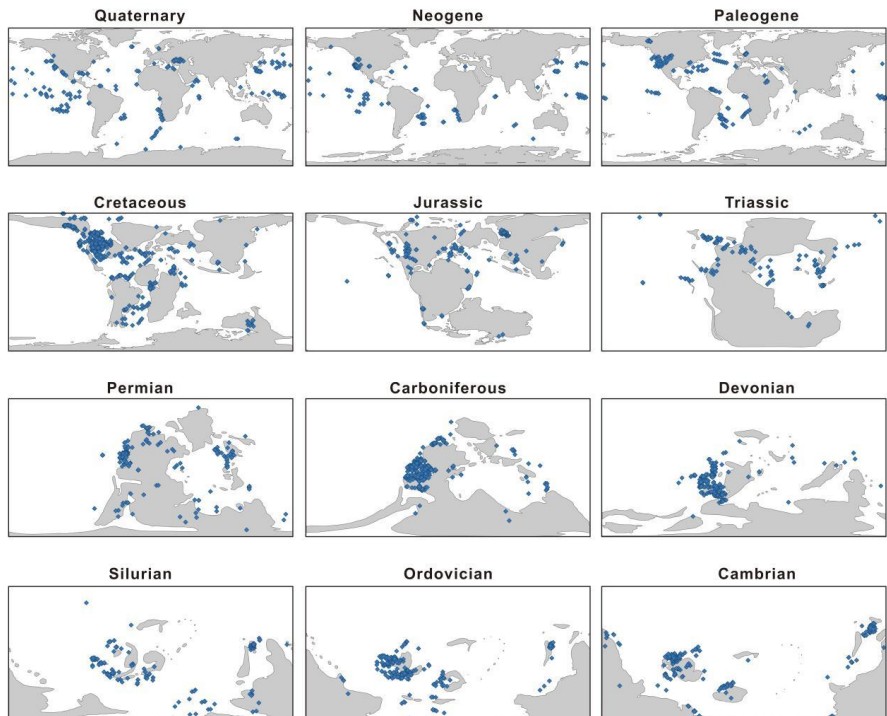


**Figure 6. The palaeogeographic distribution of sample sites in the DM-SED.**

When averaging all Phanerozoic data by stage and spatially averaging them into 15°

palaeolatitude bins (Fig. 7), Palaeozoic data records are mainly biased toward tropical regions.
Cambrian data are concentrated between 15° S and 30° N, Ordovician to Carboniferous data are
concentrated between 45° S and 15° N, and Permian data are concentrated between 0° N and 30° N,
with data mainly fluctuating around the equator. As continents migrated northward through the
Mesozoic and into the Cenozoic, records began to show bias toward mid-latitudes in the Northern
Hemisphere. From the Triassic to the Cretaceous, data are mainly concentrated between 0° N and 60°
N.

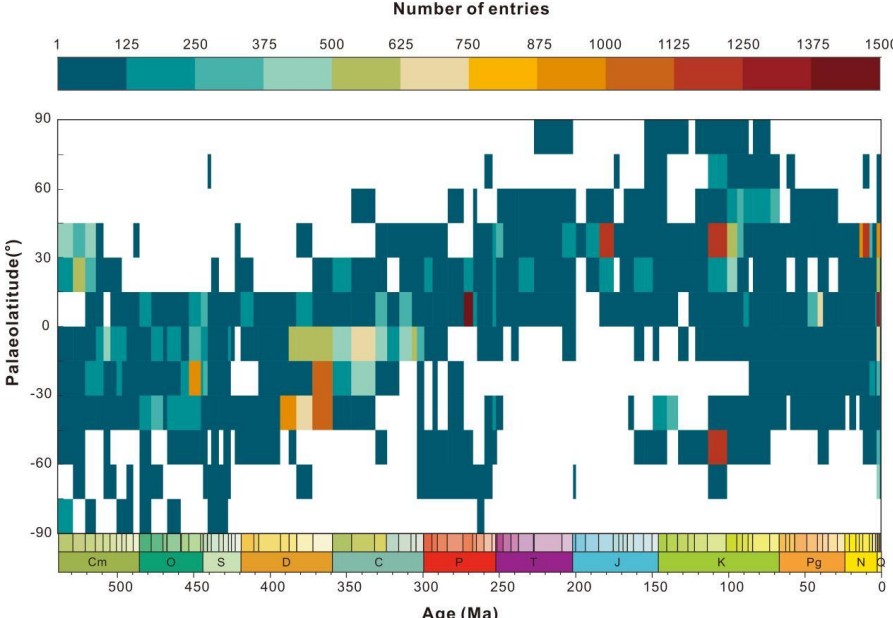


**Figure 7. The spatiotemporal distributions of sample quantities (categorized temporally by stage and**

**spatially by palaeolatitude intervals of 15°).**


**5 Usage instructions**
The ultimate goal of the DM-SED database is to provide the geoscience community with a valuable
resource of knowledge and geographic information. By deriving meaningful conclusions from a large
marine sediment geochemistry dataset, we aim to enhance our understanding of Earth's environmental
changes over time and space. All entries in DM-SED contain the source of original proxy values,
ensuring traceability between DM-SED and the original datasets from which the data were extracted.

However, our database has some limitations. The criteria for age determination, relying variously

on fossil zones and lithostratigraphic unit information, are not entirely uniform. Some age
determinations are still coarse, with samples from a single section were assigned the same age.
Additionally, the data quantity for some elements is still low. The testing methods for elements are not
annotated, and there may be significant differences in methodological precision between older and
newer literature. Currently, these issues remain largely unresolved. Despite our best efforts to identify



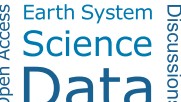

data from the literature and process quality control for each entry, the sheer volume of data in DM-SED
means that some errors or omissions are inevitable. Prompt corrections and continuous updates are
expected to ensure the credibility of this dataset.

Finally, it is important to recognize that DM-SED merely compiles these various datasets and

cannot impose any requirements on their generation. When using the data (and where practicable), we
recommend citing both DM-SED and the original data sources to ensure proper attribution.

**6 Data availability**

Version    controlled    releases    of    the    DM-SED    can    be    found    on    Zenodo
(https://doi.org/10.5281/zenodo.13898366, last accessed: 7 October 2024) (Lai et al., 2024). A static
copy    of    DM-SED    version    0.0.1    is    archived    in    the    Geobiology    data
(http://202.114.198.132/dmgeo-geobiology-portal/, last accessed: 25 September 2024). We plan to
supplement and improve the dataset continuously and hope to collaborate with existing compilation
authors to assist in adding new content.

**7 Code availability**

The software tools used in this study are available at the following links: WebPlotDigitizer can be
downloaded from https://github.com/automeris-io/WebPlotDigitizer/releases (last accessed: 20 July
2024); the PointTracker v7 tool can be found at http://www.paleogis.com (last accessed: 20 July 2024);
QGIS 3.16 can be downloaded from the https://qgis.org/project/overview/ (last accessed: 20 July

2024).


**Author contributions.** Jiankang Lai: Writing – original draft, Visualization, Data collection,
Investigation. Haijun Song: Writing – review & editing, Supervision, Investigation, Funding
acquisition. Daoliang Chu: Writing – review & editing, Investigation. Jacopo Dal Corso: Writing –
review & editing, Investigation. Erik A. Sperling: Writing – review & editing, Investigation.
Yuyang Wu: Writing – review & editing, Supervision, Investigation, Data collection. Xiaokang Liu:



Writing– review & editing, Investigation. Lai Wei: Writing– review & editing, Data collection,
Investigation. Mingtao Li: Writing– review & editing, Investigation. Hanchen Song: Writing– review
& editing, Investigation. Yong Du: Writing– review & editing, Investigation. Enhao Jia: Writing–
review & editing, Investigation. Yan Feng: Writing– review & editing, Investigation. Huyue Song:
Writing– review & editing, Investigation. Wenchao Yu: Writing– review & editing, Investigation.
Qingzhong Liang: Writing– review & editing, Investigation. Xinchuan Li: Writing– review & editing,
Investigation. Hong Yao: Writing– review & editing, Investigation.

**Competing interests.** The authors declare that they have no conflicts of interest.

**Acknowledgements.**
We thank Xiang Shu for the discussions on analytical methods. This paper benefited greatly from
comments from xxx anonymous reviewers.

**Funding:**
This study was supported by the National Natural Science Foundation of China grant 42325202, the
State Key R&D Project of China (2023YFF0804000), 111 Project grant B08030, and Natural Science
Foundation of Hubei (2023AFA006). E.A.S. is supported by United States National Science
Foundation grant EAR-2143164.

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
