# Peer review of "Deep-Time Marine Sedimentary Element Database"

_Earth System Science Data, 2024_

## Author Response (AR1)

**Note: original reviewer comments in black; responses of authors in blue**

Reviewer #1

**General comment**

Lai and co-authors present an extensive database on geochemical data (major and trace elements, stable isotopes) from deep-time (3.8 Ga to present) marine sediments, i.e. the Deep-Time Marine Sedimentary Element Database (DM-SED). The DM-SED builds upon the Sedimentary Geochemistry and Paleoenvironments Project (SGP) previously presented by Farrel et al. (2021), which is here significantly expanded adding nearly 35,000 new entries, totaling 63,691 entries with 2,412,085 discrete data points. New data supplement spatiotemporal gaps in the SGP and are largely from the Phanerozoic, although substantial new contribution for the Precambrian is provided as well.

I commend Lai et al. for digging out so many new data from the literature. The paper reads well, is clear and concise, and overall sufficient detail is provided in each section. The database is relatively well organized and easy to read. I appreciated that the authors made clear the limitations of their database, which my hamper the use of some listed entries. About this point, my concern is that the lack of any information on the methodology used to obtain the data compiled in DM-SED could discourage some users, given that methodological precision may vary significantly for elements. Therefore, I encourage the authors to integrate this information when available. Alternatively, they could add one or more entries to provide information on the methodology at a later time while keeping the database up to date.

Concluding, I would recommend this database and the companion paper for publication after minor revisions, as they provide a valuable contribution to help geoscientists aiming to improve our knowledge on climate, environments, and biogeochemical cycles from the geological past.

Author Response: Thank you for your thoughtful comments and recognition of our effort in expanding the geochemical database. We recognize the importance of sharing methodological details to enhance transparency and user confidence. To address this, we have added three additional fields to the database to document the test methods for TOC, major elements, and trace elements (table3, lines 231-235 and dataset). For more detailed test details or isotope test methods, we recommend that readers rely on the data source to find the original paper.

**Specific comments**

Section 3 – Dataset screening and processing

For completeness, it may be useful to specify the exact coordinate reference system used to express Modern latitude and longitude.

Author Response: Revised. We have now clarified in the manuscript that latitude and longitude

are expressed using the WGS84 coordinate reference system (lines 189-190).

Database file (static copy v. 0.0.1) I recommend the authors to follow the order listed in lines 127-135 and Table 3. Specifically, I suggest to move isotopic values immediately after carbon element values, and place References and Project details on the far right. I also suggest to avoid abbreviations in the field LithName. If information on the methodology will be provided, it may be included before listing References and Project details.

Author Response: Revised. The column order in the database has been updated to match the sequence described in the manuscript (lines 145-154 and Table 3). We have replaced abbreviations in the LithName field with full terms to improve clarity (e.g. complete the abbreviation "argl. mudstone" to "argillaceous mudstone" ). A new "Methodology" field has been added before the References and Project details columns to address this recommendation.

**Technical Comments**

Line 18: I suggest to replace 'for studying' with 'to study'
Author Response: Replaced. This has been corrected to ensure grammatical accuracy (line 18).

Line 25: since the data set focuses only on few isotopes (O, C, S, and N), it may be worth to specify them here or maybe make clear that only stable isotopes are targeted (as opposed to radiogenic ones, for instance)
Author Response: We have revised the sentence to indicate that the database focuses only on stable isotopes (O, C, S, N) (line 25).

Line 32: technically speaking I think 'concentration' is not correct here because you are referring to a solid. I suggest content, distribution or amount
Author Response: We have replaced "concentration" with "content" for accuracy, as the reference is to a solid (line 32).

Line 34: what does 'its' refer to? please be more specific
Author Response: We have specified "its" as referring to "Earth's surface systems" (lines 33-34).

Line 36: replace 'reconstruct' with 'reconstructing'
Author Response: Replaced. We have replaced 'reconstruct' with 'reconstructing' (line 36).

Line 37: consider 'perturbations' instead of 'changes'; I would also suggest to end the sentence as '… thereby revealing mechanisms driving past climate fluctuations'
Author Response: Revised. We have adopted the suggested phrasing: "… thereby revealing mechanisms driving past climate fluctuations." (lines 37-38)

Line 43: suggested change 'Oxygen isotopes (d18O) from fossilized marine organisms'
Author Response: The sentence has been revised as suggested: "Oxygen isotopes (d18O) from fossilized marine organisms." (line 43)

Lines 84-87: consider simplifying this sentence. A suggestion may be '…data on ancient marine

sediments, they have shortcomings such as limited spatial coverage, the lack of age data and coarse age resolution, the absence of recent publications, and missing information from original publications.'

Author Response: The sentence has been simplified to: "…data on ancient marine sediments, they have shortcomings such as limited spatial coverage, lack of age data, coarse age resolution, absence of recent publications, and missing information from original publications." (lines 92-94)

Line 87: replace 'have established' with 'propose'

Author Response: Replaced. We have replace 'have established' with 'propose'. (line 94)

Line 96: replace 'data' with 'portal' or 'database'. The same goes for line 312

Author Response: Replaced. We have replaced "data" with "database" for clarity here (line 101) and in line 341.

Lines 105-107: the sentence is not clear in its current form. Please consider rephrasing

Author Response: Rephrased. The sentence has been rephrased to improve clarity. "The database is primarily sourced from the SGP database, supplemented with additional 34,874 newly compiled entries. The SGP contains a total of 82,578 entries, from which we selected 28,753 entries specifically related to marine sedimentary geochemical data,…." (lines 121-124)

Line 115: suggested change ' … entries from 433 studies, spanning approximately 3800 Ma and including entries…'

Author Response: Revised. The sentence has been updated to: "… entries from 433 studies, spanning approximately 3800 Ma and including entries…." (lines 132-133)

Lines 114-118: the sentence is quite long. It may be better to split it in two after the list of all countries

Author Response: Revised. The sentence has been split into two for improved readability. "Our DM-SED database, built upon the SGP, includes a new compilation of 34,874 entries from 433 studies, spanning approximately 3800 Ma and including entries from North America, Europe, Asia, Africa, South America, Oceania, Pacific and Atlantic. This supplements the temporal and spatial distribution gaps in the SGP database, thereby creating a more comprehensive sedimentary marine geochemical database." (lines 131-135)

Line 131: the comma after TOC is probably a mistake and there is a bracket missing here

Author Response: The typo has been corrected. (line 149)

Line 192: suggested change 'types of rocks' or 'rock types'

Author Response: We have replaced "types of rocks" for consistency. (line 216)

Line 193: add 'the' before lithostratigraphic

Author Response: Added. The suggested addition has been made. (line 217)

Line 194: maybe you could rephrase as 'however, for data from marine drilling sites'. What do you mean exactly with 'there were no corresponding group names'? I this referred to lithological information? please be more precise here

Author Response: Rephrased. We have clarified the sentence to specify that "no corresponding group names" refers to missing lithological information. (lines 218-219)

Lines 209-210: 'And the project includes two parts: new compilation and SGP' please rephrase this sentence

Author Response: Rephrased. This sentence has been rephrased for better clarity. "The entire database for this Project was divided into two parts: new compilation and SGP" (lines 239-240)

Line 223-224: suggested change 'For carbon elements, TOC has the largest record (33,216 entries), followed by Total C (9,201 entries), while Cinorg has the lowest record (7,194 entries).'

Author Response: Revised. The text has been revised as suggested: "For carbon elements, TOC has the largest record (32,904 entries), followed by Total C (9,386 entries), while $C_{inorg}$ has the lowest record (7,215 entries)." (lines 253-254)

Line 232: be more specific here. I suggest 'Within the Phanerozoic' in place of 'From this'

Author Response: Revised following your suggestion. (lines 262-263)

Lines 231-239: it may be good her to provide some temporal references. For instance age intervals for the Phanerozoic and Proterozoic Eons, and for the main Eras mentioned in the text. Alternatively, my suggestion is to include Eons in Figure 4a and Eras in Figure 4b (see below)

Author Response: Revised. Figures 4a and 4b have been updated to include Eons and Eras for improved clarity.

Line 285: I think adding a short last sentence on the distribution of data in the Ternary-Quaternary may be useful to increase completeness of the description

Author Response: Added. A short description of data distribution in the Ternary-Quaternary has been added to increase completeness. "Paleogene to Quaternary data are concentrated between 45° S and 45° N". (line 315)

Figure 3: is there a specific reason why the list in each table is in reverse order than mentioned in the text (lines 130-135)?

Author Response: The order of items in Figure 3 has been updated to match the sequence mentioned in the text. (lines 148-154)

Figure 4: I suggest the authors to indicate here Eons (Fig. 4a) and Eras (Fig. 4b) mentioned in the text to improve clarity. Additionally, abbreviations should be clarified, if not done elsewhere

Author Response: Revised. We have updated Figure 4 to include labels for Eons (Figure 4a) and Eras (Figure 4b) and ensured all abbreviations are clarified in the figure caption.

Thank you for your detailed and constructive feedback. We believe these changes have significantly improved the manuscript and database.

Reviewer #2    Thierry Adatte

Lines 62-92:The authors may also include the NOAA and MMS Marine Minerals Geochemical Database. This database contains geochemical analyses and auxiliary information on present-day marine deposits, primarily ferromanganese nodules and crusts, as well as some data on heavy minerals and phosphorites. It is maintained by the National Centers for Environmental Information (NCEI).

They also may include GEOTRACES, which provides hydrographical and marine geochemical data acquired over the past decade. The data covers the global ocean, with a focus on trace elements and their isotopes, aiding in the study of marine biogeochemical cycles.

Author Response: Thank you for the suggestion. We have included the NOAA and MMS Marine Minerals Geochemical Database and GEOTRACES in the text (lines 73-80) and Table 1.

Lines 131-132:There are many more exotic isotopes that could be useful to add to the DM-SED, such as:

$\delta 98/95$Mo (Molybdenum isotopes) which indicates redox conditions, particularly euxinia (anoxic and sulfidic conditions) in marine environments.

$\delta 238/235$U (Uranium isotopes), which tracks redox conditions and changes in seawater chemistry.

$\delta 44/40$Ca (Calcium isotopes), which tracks carbonate precipitation and dissolution, diagenesis, and biological activity.

$\delta 30$Si (Silicon isotopes), which Indicates biogenic silica production and nutrient cycling.

$\delta 114/110$Cd (Cadmium isotopes) which reflects nutrient utilization and past productivity in surface waters.

$\delta 54/52$Cr (Chromium isotopes) Tracks redox-sensitive processes and oxidative weathering.

$87Sr/86Sr$ (Strontium isotopes) which reflects changes in seawater composition due to continental weathering and hydrothermal activity.

$143Nd/144Nd$ (Neodymium isotopes) which tracks water mass mixing and provenance of sediments.

$187Os/188Os$ (Osmium isotopes) which indicates continental weathering rates and extraterrestrial inputs.

Δ47 Clumped Isotopes must be also added since they provide direct estimates of past temperatures independent of seawater δ18O.

Author Response: We appreciate your suggestion to include additional isotopic data. However, at this stage, our dataset focuses mainly on element data. And stable isotope data in current version of DM-SED database is limited and not the focus of the study. Additionally, these isotopes often require specialized analytical techniques, and comprehensive datasets are scarce in the published literature. We will consider adding such data in future updates as relevant datasets become more widely available.

Line 144-212:Data set screening I don't know if it is too difficult, but it would be good to roughly indicate the methods which has been used.

Author Response: Added. We have now included a general description of the methods commonly used for TOC, major elements, and trace elements in the manuscript (table3, lines 231-235). Additionally, where available, methodological details have been added to the database to provide more transparency.

Line 189:For sure that GTS 2020 remains highly accurate for most applications, but it's essential to stay informed of incremental updates in absolute ages and boundary definitions, especially for precise or controversial stratigraphic intervals. International Chronostratigraphic Chart (ICS) online for the latest updates (ICS website).

Author Response: Added. We agree with this point and have added a note in the manuscript emphasizing the importance of consulting the latest updates from the International Chronostratigraphic Chart (ICS) for precise or controversial stratigraphic intervals. "Although GTS 2020 is accurate, readers are advised to consult the incremental updates of the International Chronostratigraphic Chart (ICS) for the most accurate stratigraphic intervals". (lines 212-214)

Line 191"dolomite", the authors may use dolostone and not dolomite, which is a mineral and not a lithology sensu stricto

Author Response: Replaced. We have replaced "dolomite" with "dolostone" throughout the manuscript to ensure accuracy. (line 215)

Line 192:This classification is a bit too large. It would be possible to be more precise ?

Calcareous Rock\Marls (35-60 % of carbonate)\Sand\Clay

Author Response: We understand your concern, but given the diverse nature of the compiled dataset, applying a more detailed classification scheme may not be consistent or feasible across all entries. To maintain simplicity and usability, we have retained the broader classification. However, we will consider refining the classification in future updates where additional metadata allows.

Line 196:The authors may include the tidal flats into the inner shelf subdivision.

Author response: Revised. We agree that tidal flats should be added to the inner shelf subdivision, so we have deleted tidal flats from the original paper. (line 220)

Line 197:Ok, but there are also many data set coming from shallow settings.

Author response: Revised. We acknowledge that many data after the Cretaceous also come from shallow settings, so we changed the original text to: "However, after the Cretaceous, with most samples coming from the DSDP and ODP, shallow marine depositional environments still existed and were sampled, but deep-sea pelagic settings began to be sampled as well."(lines 221-223)

Line 231-241 and figure 4:The number of entries for the 100-113 Ma interval (Albian) is surprisingly high compared to the 100-90 Ma interval (Cenomanian), which includes the OA2 event, one of the best studied intervals of the Cretaceous. The low number of Paleogene entries is also striking, especially for the Lower Paleogene, which includes several hyperthermals (including the PETM) that are among the most studied events of the Phanerozoic.

Author Response: Thank you for your detailed feedback. We have verified the data distribution and confirmed that the higher number of Albian entries reflects biases in published studies. The lower representation of Paleogene entries is likely due to gaps in available data, which we aim to address in future updates.

In Figure 4a, it would be good to include eons. Furthermore, it is not sure that the potential readers are familiar enough with the Proterozoic periods to understand the abbreviations used (e.g. Ec (Ectasien)). There is also a problem of scale in figure 4b: the rectangle corresponding to the Albian is almost twice as wide as that of the Aptian, but their duration is almost the same, around 12 million years. It would be clearer for the reader if the eras were also shown in Figure 4b.

Author Response: Revised. For Figure 4, we have included eons in Figure 4a, adjusted the scale in Figure 4b to better reflect the duration of each interval, and added eras to Figure 4b for improved readability. We also ensured all abbreviations are clarified in the figure caption (e.g. Ec (Ectasien)).

Lines 249:That sentence is difficult to understand. The Quaternary period includes the Pleistocene and Holocene stage.

Author Response: Thank you for your suggestion. Pleistocene and Holocene belong to Series rather than Stages in the Quaternary. So we emphasize here: 'the data were not subdivided by Stage but were instead divided into the Holocene and Pleistocene Series'(lines 278-279).

Lines 257:As said above, to my knowledge the OA2 is clearly more studied than the OA1b.

Author Response: Thank you for this observation. While we agree that the OA2 interval is widely studied, the database's composition reflects the availability of data from published studies rather than a bias in the database design. Therefore, we are unable to adjust the data distribution.

Lines 275:are mainly *located* in the deep ocean

Author Response: Revised. We agree with this point and have clarified the statement to ensure accuracy.(line 305)

Thank you again for your constructive feedback. We have carefully addressed your comments to improve the manuscript and database.

---

## Referee Report (RR1)

**Review of paper n. essd-2024-435 'Deep-Time Marine Sedimentary Element Database'**

General comment

Lai and co-authors have done well in addressing all review comments, further improving the credibility of their DM-SED database and the readability of the manuscript. Therefore, I would recommend this database and the companion paper for publication after few minor technical corrections.

Technical comments

Lines 273: 'Sd, Siderian' is repeated here

Line 303: I think something is missing here. In the previous version you referred to shelves. It may be worth restoring the word or clarifying this sentence